*Report*

# The NLRP1 inflammasome is an essential and selective mediator of axon pruning in neurons

Selena E Romero[1], Matthew J Geden[2], Richa Basundra[2], Kiran Kelly-Rajan [ID][2], Edward A Miao [ID][3] & Mohanish Deshmukh [ID][1,2][✉]

## Abstract

**Axon pruning is a unique process neurons utilize to selectively degenerate axon branches while keeping the neuronal cell body intact. The mechanisms of axon pruning have much in common with those of apoptosis. Both axon pruning and apoptosis pathways require key apoptotic proteins (Bax, Caspase-9, Caspase-3). Interestingly, axon pruning does not require Apaf-1, a key member of the apoptosome complex. As such, exactly how caspases are activated in an apoptosome-independent manner during axon pruning is unknown. Here we show that neurons utilize the NLRP1 inflammasome, an innate immune sensor of pathogens, specifically for axon pruning. Strikingly, NLRP1b-deficient neurons were unable to prune axons both in vitro and in vivo, but fully capable of degenerating during apoptosis. Our results reveal NLRP1 as an immune molecule engaged by neurons for an unexpected physiological function independent of its pathogen-induced proinflammatory role.**

**Keywords** Axon Pruning; Apoptosis; Inflammasome; NLRP1; Caspase-1
**Subject Categories** Autophagy & Cell Death; Neuroscience; Post-translational Modifications & Proteolysis

## Introduction

Axon pruning is the process of selectively degenerating misguided axons while maintaining the survival of the neuronal cell body (Low and Cheng, 2006; Riccomagno and Kolodkin, 2015). This process is known to occur throughout development in many types of neurons (Bagri et al, 2003; Bishop et al, 2004; O'Leary and Koester, 1993; Singh et al, 2008), and its dysregulation is implicated in numerous neurodevelopmental disorders (Riccomagno and Kolodkin, 2015; Thomas et al, 2016) and neurodegenerative diseases (Fischer et al, 2004; Kanaan et al, 2013; Stokin et al, 2005). The mechanisms of axon pruning have been well studied using nerve growth factor (NGF) deprivation of peripheral neurons, specifically in compartmentalized culture systems such as

microfluidic chambers (Cusack et al, 2013; Maor-Nof et al, 2016). These microfluidic chambers allow for both the spatial and fluidic isolation of neuronal soma from their distal axons. This is a particularly important feature as NGF deprivation in different contexts results in two distinct outcomes: apoptosis or axon pruning. Removal of NGF from both the soma and axons (global deprivation) activates the apoptotic pathway, resulting in the degeneration of the entire neuron (Geden et al, 2019). In contrast, the deprivation of NGF solely from distal axons induces the axon pruning pathway where only the targeted axons degenerate (Geden et al, 2019).

The pathways of apoptosis and axon pruning in peripheral neurons are similar, but with striking differences. In both contexts, NGF deprivation activates a c-Jun N-terminal kinase (JNK) signaling pathway (Ghosh et al, 2011; Simon et al, 2016) resulting in the phosphorylation and activation of the transcription factor c-Jun. This induces the expression of select BH3-only proteins that facilitate the activation of the Bax, a well characterized apoptotic protein. Bax, once activated, translocates to the mitochondria and forms pores in the outer mitochondrial membrane, releasing cytochrome $c$ (cyt $c$) from the mitochondria into the cytosol. In the context of apoptosis, cytosolic cyt $c$ oligomerizes with Apaf-1 and Procaspase-9 (Casp9) to form the apoptosome complex which activates Casp9 and Caspase-3 (Casp3), and ultimately causes the programmed death of the neuron (Hollville et al, 2019). Axon pruning, however, is surprisingly not dependent on Apaf-1 despite requiring both Casp9 and Casp3. Interestingly, axon pruning also requires an additional caspase, Caspase-6 (Casp6), which is not essential for apoptosis (Cusack et al, 2013). While axon pruning requires these three caspases, exactly how they are activated in an apoptosome-independent manner is unknown.

In non-neuronal cells one mechanism by which caspases can be activated independently of the apoptosome is through inflammasome protein complexes (inflammasomes) (Barnett et al, 2023; Nozaki et al, 2022a). Inflammasomes, part of the innate immune system, form in response to specific pathogens or catastrophic cellular damage (Barnett et al, 2023; Man and Kanneganti, 2016; Nozaki et al, 2022b). These various triggers are recognized by different sensor proteins which, in conjunction with the adapter protein ASC (Apoptosis-associated Speck-like protein containing a CARD), activate the inflammatory caspase, Caspase-1 (Casp1) (Barnett et al, 2023).

---

[1]Department of Cell Biology and Physiology, University of North Carolina, Chapel Hill, NC, USA. [2]Neuroscience Center, University of North Carolina, Chapel Hill, NC, USA. [3]Department of Integrative Immunobiology, Duke University, Chapel Hill, NC, USA. [✉]E-mail: mohanish@med.unc.edu

   

While inflammasomes have been primarily studied in the context of immune cells, neurons interestingly also express many of these inflammasome components (Kaushal et al, 2015; McKenzie et al, 2020). However, it is unclear what function inflammasomes have specifically within neurons. Studies have shown that Casp1 is capable of activating Casp6 in biochemical assays (Guo et al, 2006), and in models of Alzheimer's Disease (AD) in human cortical neurons (Kaushal et al, 2015). Whether inflammasomes are required for the maintenance of neuronal homeostasis, such as in axon pruning, is unknown. In this study we report an unexpected function of the NLRP1/Casp1 inflammasome in neurons where they are essential mediators of axon degeneration selectively during physiological axon pruning but not apoptosis.

# Results and discussion

## Caspase-1 is required for axon pruning

To probe alternate mechanisms of caspase activation during axon pruning, we first examined whether Casp1 was required in this context using the established in vitro microfluidic chamber model. Specifically, axon pruning was induced by the removal of nerve growth factor (NGF) from the distal axons of sympathetic neurons grown in compartmentalized cultures, causing the selective degeneration of targeted axons (Fig. 1A). Neurons were treated with the Casp1 specific inhibitor Ac-YVAD-cmk to assess the requirement of Casp1 in pruning. Importantly, Ac-YVAD-cmk has no effect on the other inflammatory caspase, Caspase-11 (Kang et al, 2002). Casp1 inhibition strikingly reduced axon degeneration during pruning (Fig. 1B,C). Additionally, Casp1 was not important for axon degeneration in general as Casp1 inhibition did not block degeneration during apoptosis (Fig. 1B,C).

To confirm the importance of Casp1 and rule out Casp11, we examined axon pruning in neurons that were deficient for both Casp1 and Casp11 (CASP1/11−/−), or just deficient for Casp1 (CASP1−/−). Both the CASP1/11−/− mice and the CASP1−/− mice have been widely used to study the function of Casp1 in the context of inflammasome activation (Lammert et al, 2020; Okondo et al, 2017; Rauch et al, 2017). CASP1/11 deficiency inhibited axon degeneration during pruning, but not during apoptosis. Importantly, Casp1 deficiency alone was similarly effective in preventing axon pruning, with less than 5% of axons degenerated in Casp1-deficient neurons after axonal NGF deprivation (Fig. 1D,E). These results indicate that Casp1 is essential for axon pruning. As anticipated, Casp1 deficiency did not block axon degeneration during apoptosis, highlighting the differences in the mechanisms of axon degeneration during pruning and apoptosis (Fig. 1D,E).

## The NLRP1 inflammasome mediates axon pruning

In the context of the innate immune system, Casp1 is activated by inflammasomes (Fig. 2A) (Barnett et al, 2023; Man and Kanneganti, 2016; Nozaki et al, 2022a). As we found Casp1 to be required for axon pruning, we examined whether neurons express key inflammasome components. Interestingly, we found that sympathetic neurons do indeed express many of the well characterized inflammasome components, including the inflammasome sensors

Absent in Melanoma 2 (AIM2), the Nucleotide-binding domain and Leucine-rich Repeat containing Proteins 1 and 3 (NLRP1, NLRP3), as well as the key adapter, ASC (Fig. 2B).

We next asked whether inflammasomes were required for the non-pathogenic process of axon pruning. First, we examined the inflammasome sensor AIM2 which can be activated by cytosolic mitochondrial DNA (Newman and Shadel, 2023). AIM2 deficient neurons exhibited similar axon degeneration as the wild-type controls indicating that AIM2 was not required for axon pruning (Fig. 2C,D).

Another family of inflammasome sensor proteins is the NLR family which include NLRP1 and NLRP3. NLRP3 in particular has been very well characterized and can be activated by K+ flux, mitochondrial reactive oxygen species (mtROS), changes in ATP, and mitochondrial antiviral signaling protein (MAVS) (Barnett et al, 2023; He et al, 2016; Nozaki et al, 2022a). As NLRP3 could be activated during pruning by changing ATP levels or by mtROS (Barnett et al, 2023), we next examined whether NLRP3 was required in this pathway. However, NLRP3 deficient neurons also exhibited normal axon pruning (Fig. 2C,D). Because both the AIM2 and NLRP3 sensor proteins were not required, we instead focused on ASC, an essential adapter protein for most inflammasomes to bind and activate Casp1 (Man and Kanneganti, 2016; Nozaki et al, 2022a). Neurons deficient in ASC however did not exhibit any axon pruning defects, indicating the adapter was also not necessary for pruning (Fig. 2C,D). Thus, the most common canonical inflammasome components (i.e., AIM2, NLRP3, ASC) did not appear to be required for axon pruning.

Interestingly, NLRP1 can directly activate Casp1 independently of ASC via its caspase activation and recruitment domain (CARD) (Kovacs and Miao, 2017; Man and Kanneganti, 2016). Mice have three paralogs of NLRP1 (NLRP1a, b, and c) with NLRP1b being the best defined (Chavarria-Smith and Vance, 2015; Sastalla et al, 2013). We examined if NLRP1b was required for axon pruning using neurons isolated from NLRP1b-deficient mice (Kovarova et al, 2012). Strikingly, NLRP1b deficiency completely blocked axon degeneration during pruning with axons remaining continuous and intact, even after 96 h of axonal NGF deprivation (Fig. 2E,F). In contrast, NLRP1b-deficient neurons fully degenerated during apoptosis (Fig. 2E,F).

To confirm that the effect observed in the NLRP1b−/− mice was not the result of defects in other inflammasome machinery, we examined the expression of *CASP1*, *AIM2*, *NLRP3*, and *ASC* in the NLRP1b-deficient sympathetic neurons. These inflammasome components were expressed at similar levels in the NLRP1b-deficient neurons as compared to wild-type (Fig. EV1). Together, our results identify the inflammasome sensor NLRP1b as an essential component selectively of the axon pruning pathway.

## NLRP1 functions downstream of c-Jun phosphorylation

To establish if NLRP1 activation is an early or late event during pruning we focused on the phosphorylation of c-Jun. c-Jun phosphorylation is one of the earliest observable events in both the apoptosis and axon pruning pathways (Geden et al, 2019; Ghosh et al, 2011). As anticipated, both global NGF deprivation (apoptosis) and axonal NGF deprivation (axon pruning) induced the phosphorylation of c-Jun which was visible in the nuclei (Fig. 3A,B). The percentage of phospho-c-Jun positive neurons

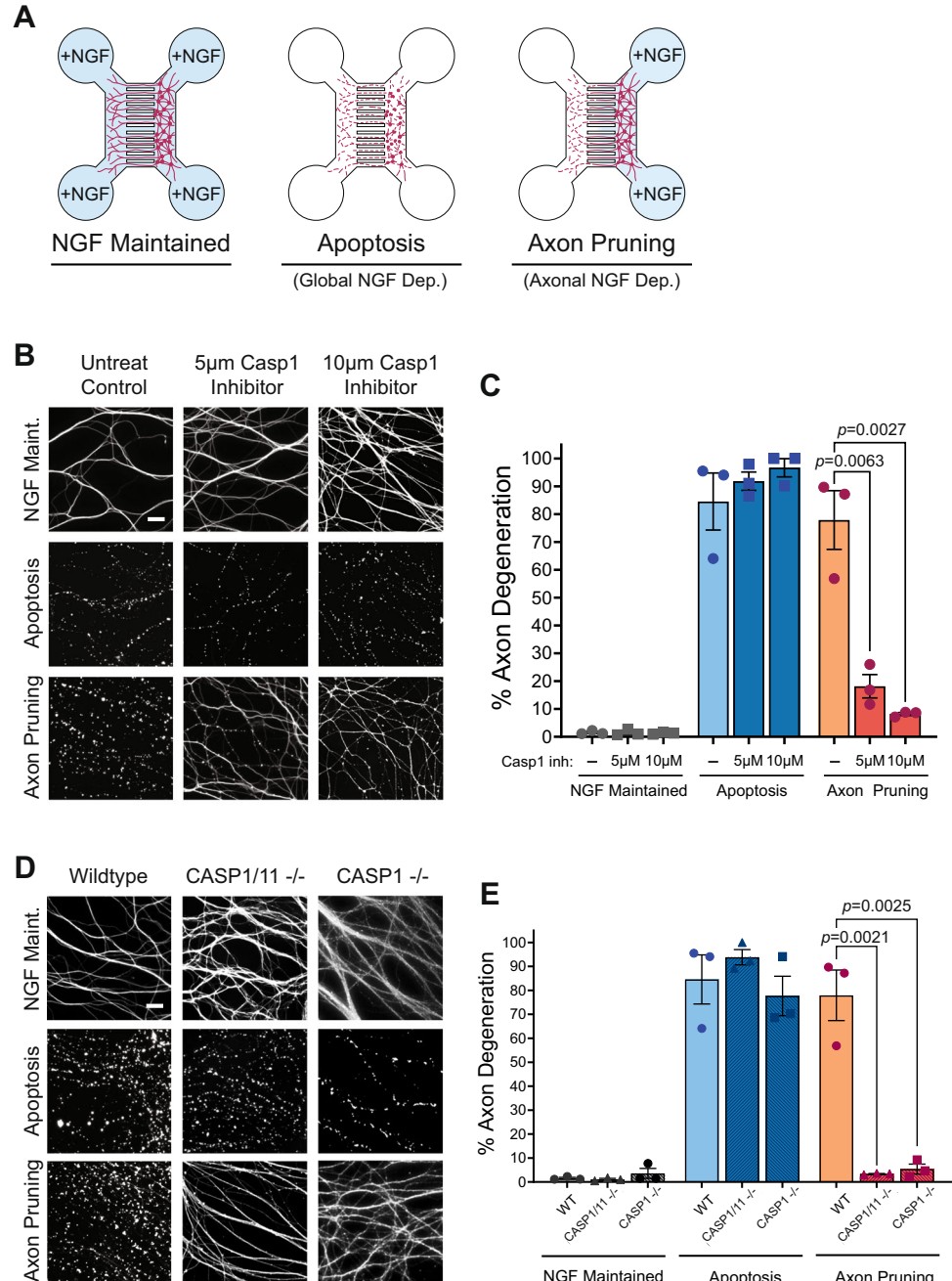

**Figure 1. Caspase-1 inhibition or deficiency prevents axon degeneration specifically during axon pruning.**

(A) Schematic of microfluidic chambers utilized to culture neurons, induce apoptosis (Global NGF deprivation), or axon pruning (axonal NGF deprivation).
(B) Representative immunofluorescence images of axons (stained with α-tubulin) in the axonal compartment of microfluidic chambers in NGF maintained, apoptosis (48 h), or axon pruning (96 h) conditions treated with 5 μM and 10 μM Caspase-1 inhibitor (Ac-YVAD-cmk). Scale bar = 20 μm. (C) Percent axon degeneration quantified for experiments in (B) (n = 3 biological replicates). Individual data points and mean with standard error of mean represented. Statistical differences were examined by unpaired Student's t test. (D) Representative axonal (α-tubulin immunofluorescence) images of neurons isolated from wild-type, Caspase-1/11 deficient, and Caspase-1 deficient mice cultured in microfluidic chambers. Neurons were cultured in NGF maintained, apoptosis (48 h), or axon pruning (96 h) conditions. Scale bar = 20 μm.
(E) Percent axon degeneration quantified for experiments in (D) (n = 3 biological replicates). Individual data points and mean with standard error of mean represented. Statistical differences were examined by unpaired Student's t test. Source data are available online for this figure.

differs between apoptosis and axon pruning, because while every neuron is affected by global NGF deprivation, only the neurons that have extended their axons into the axon compartment are affected with axonal NGF deprivation. Importantly, NLRP1b-deficient

neurons displayed similar numbers of phospho-c-Jun positive nuclei after axonal NGF deprivation, suggesting NLRP1 functions downstream of (or in parallel to) c-Jun phosphorylation in the pruning pathway.

    

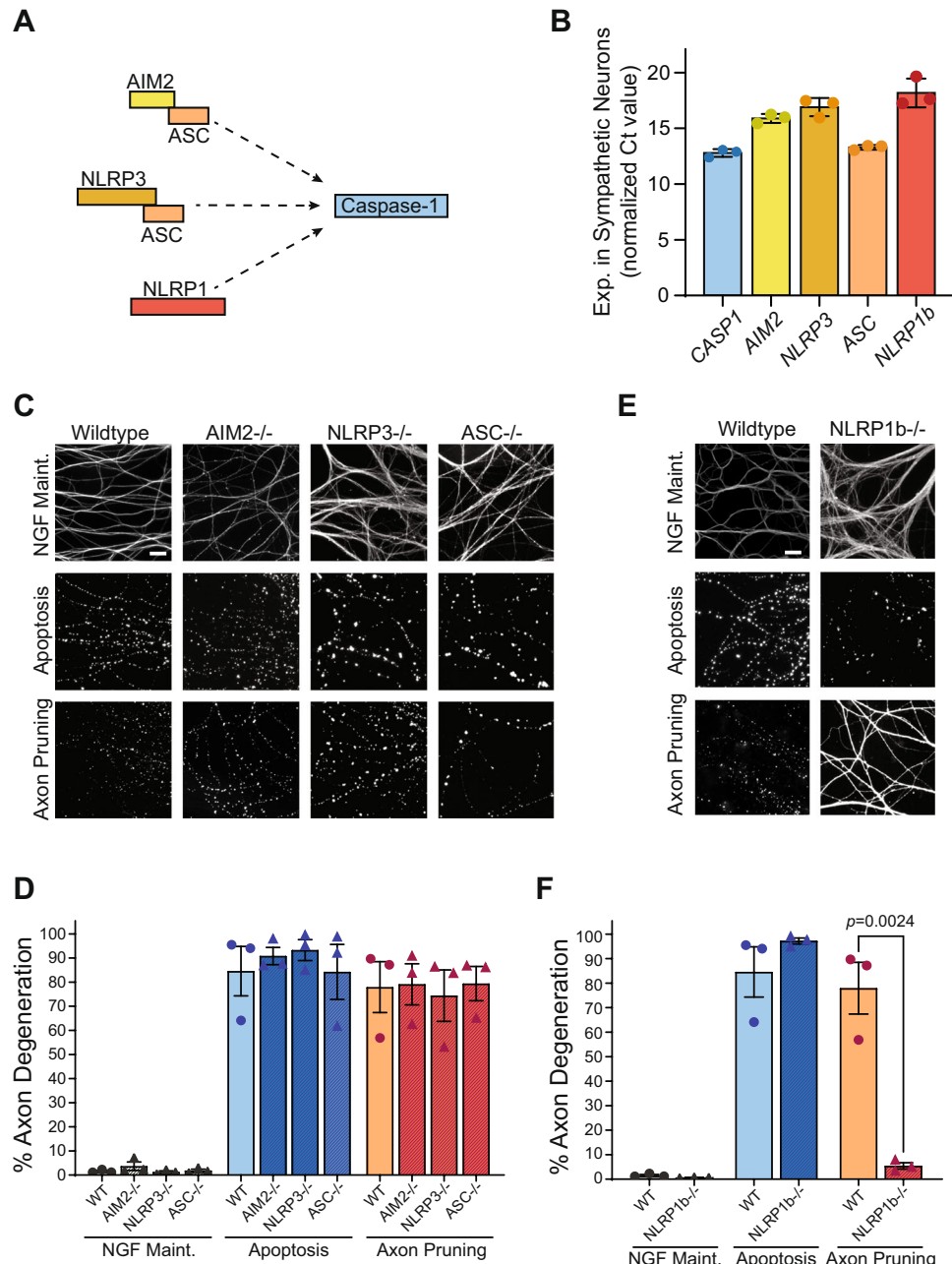

**Figure 2. The NLRP1 inflammasome specifically is required for axon pruning, not apoptosis.**

(A) Schematic of common inflammasomes known to activate Caspase-1. (B) Expression of inflammasome components *CASP1, AIM2, NLRP3, ASC, NLRP1b* in sympathetic neurons, with Ct value normalized to GAPDH. Error bars represent standard error of mean ($n = 3$ biological replicates) (C) Representative axonal (α-tubulin immunofluorescence) images of neurons isolated from wild-type, AIM2, NLRP3, or ASC deficient mice cultured in microfluidic chambers. Neurons were cultured in NGF maintained, apoptosis (48 h), or axon pruning (96 h) conditions. Scale bar = 20 μm. (D) Quantification of axon degeneration of experiments in (C) ($n = 3$ biological replicates). Individual data points and mean with standard error of mean represented. Statistical differences were examined by unpaired Student's *t* test. (E) Representative axonal (α-tubulin immunofluorescence) images of neurons isolated from wild-type or NLRP1b-deficient mice cultured in microfluidic chambers. Neurons were cultured in NGF maintained, apoptosis (48 h), or axon pruning (96 h) conditions. Scale bar = 20 μm. (F) Quantification of axon degeneration of experiments in (E) ($n = 3$ biological replicates). Individual data points and mean with standard error of mean represented. Statistical differences were examined by unpaired Student's *t* test. Source data are available online for this figure.

## NLRP1 activation alone is insufficient to induce axon or soma degeneration

We next examined whether NLRP1 activation alone in neurons is sufficient to induce axon pruning. In several cell types, pharmacological activation of endogenous NLRP1 can be achieved by treatment with Val-boroPro (VbP, Talabostat) (Okondo et al, 2017). Specifically, VbP functions to inhibit dipeptidyl peptidase 8 (DPP8) and dipeptidyl peptidase 9 (DPP9) which are known to bind and inactivate NLRP1. Inhibiting these DPPs leads to

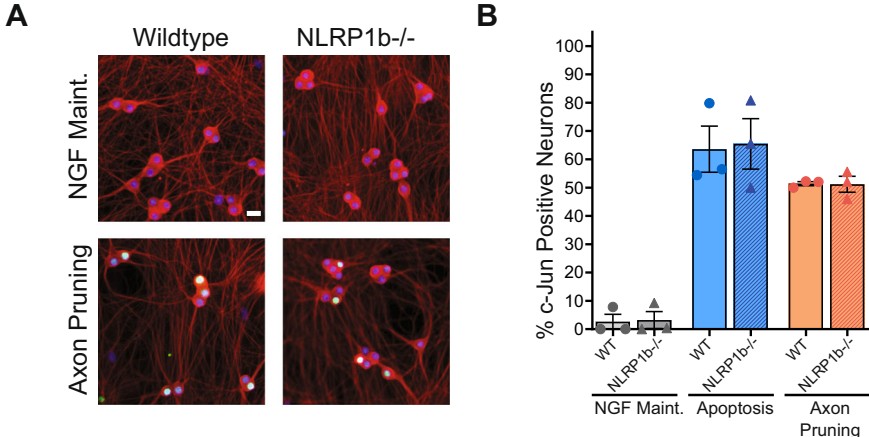

**Figure 3. NLRP1 functions downstream of c-Jun phosphorylation in the axon pruning pathway.**

(A) Representative images in the soma compartment of wild-type and NLRP1b-deficient neurons in NGF maintained and Axon Pruning conditions for 24 h. Neurons stained for tubulin (red), Hoechst (blue), and phospho-c-Jun (green). Scale bar = 20 µm. (B) Quantification of the percent of neurons in NGF Maintained, Apoptosis, or Axon pruning microfluidic chambers which are phospho-c-Jun positive (n = 3 biological replicates). Statistical differences were examined by unpaired Student's t test. Source data are available online for this figure.

spontaneous NLRP1 activation and death in immune cells (Okondo et al, 2017). We treated neurons with Vbp to induce spontaneous NLRP1 activation and found that addition of VbP to neurons, even at 5× higher concentration than required in immune cells, was insufficient to induce axon degeneration (Fig. 4A,B).

To further examine whether direct activation of NLRP1 is sufficient to induce axon degeneration, we overexpressed a constitutively active fragment of NLRP1b in neurons. This constitutively active fragment of NLRP1b, comprised of UPA-CARD domains, is released via proteasome mediated degradation and is essential for NLRP1 activation (D'Osualdo et al, 2011). Utilizing a doxycycline inducible system, we overexpressed the constitutively active UPA-CARD fragment of NLRP1b in neurons (Fig. 4C). We found that overexpression of active NLRP1b alone also did not induce neuronal degeneration (Fig. 4C,D). Thus, both our VbP and our UPA-CARD overexpression results show that while NLRP1b is required for axon pruning, its activation alone was insufficient to cause axon degeneration or cell death in neurons.

As neither inhibition of DPP8/9 via VbP nor overexpression of the UPA-CARD fragment of NLRP1b induced degeneration, we examined the outcome of combining both these treatments. Combining UPA-CARD expression in addition to VbP treatment did not increase neuronal degeneration (Fig. 4E). Interestingly however, we observed a significant reduction in soma size as compared to neurons that did not receive treatment, or neurons that received just one treatment alone (Fig. 4F). Thus, while unable to induce death, NLRP1b overexpression in conjunction with VbP is sufficient to induce neuronal atrophy, a likely consequence of neuronal stress.

## NLRP1 is required for in vivo axon pruning of the Infrapyramidal Bundle

To confirm the significance of our in vitro results, we examined whether NLRP1b was required for physiological axon pruning

in vivo. Specifically, we assessed central nervous system (CNS) axon pruning in the Infrapyramidal Bundle (IPB) of the adult mouse hippocampus (Fig. 5A) (Bagri et al, 2003). Early in development, the IPB extends nearly to the end of the SPB (suprapyramidal bundle) but is progressively pruned shorter to its final length established in adulthood (Bagri et al, 2003). Thus, axon pruning defects are expected to result in a higher IPB/SPB ratio in adult animals. Indeed, we found that the NLRP1b-deficient mice had a significantly longer IPB than wild-type litter mates and an increased IPB/SPB ratio (Fig. 5B). These findings complement our in vitro results with peripheral neurons and demonstrate that NLRP1b is required for physiological axon pruning in vivo.

Our results have identified an essential function of the inflammasome in neurons in the non-pathogenic context of axon pruning. Rather unexpectedly, axon pruning was not mediated by AIM2, NLRP3, or ASC but instead was completely dependent on the NLRP1/Casp1 inflammasome. Additionally, the NLRP1/Casp1 inflammasome was uniquely required for axon degeneration during pruning but not apoptosis. NLRP1 was also required for axon pruning in vivo as mice deficient in NLRP1b exhibited axon pruning defects within the IPB. Taken together, these results identify an unexpected physiological role of NLRP1 in neurons.

An exciting development within the field has been the recognition that immune molecules play important roles within neurons for homeostasis. For example, AIM2 recognizes DNA damage in neurons with high replicative stress and causes their developmental death (Lammert et al, 2020). Additionally, neurons utilize the complement system, specifically C1q and C3, for synapse elimination in the developing brain (Stevens et al, 2007). Immune molecules have also been of keen interest in pathological contexts. For example, significant inflammasome expression (i.e., NLRP1, NLRP3) has been reported in Alzheimer's disease (AD) brains, though this expression has been largely attributed to microglia (Freeman and Ting, 2016; Zhou et al, 2023a). However, recent work has suggested that *neuronal* inflammasome expression may play a

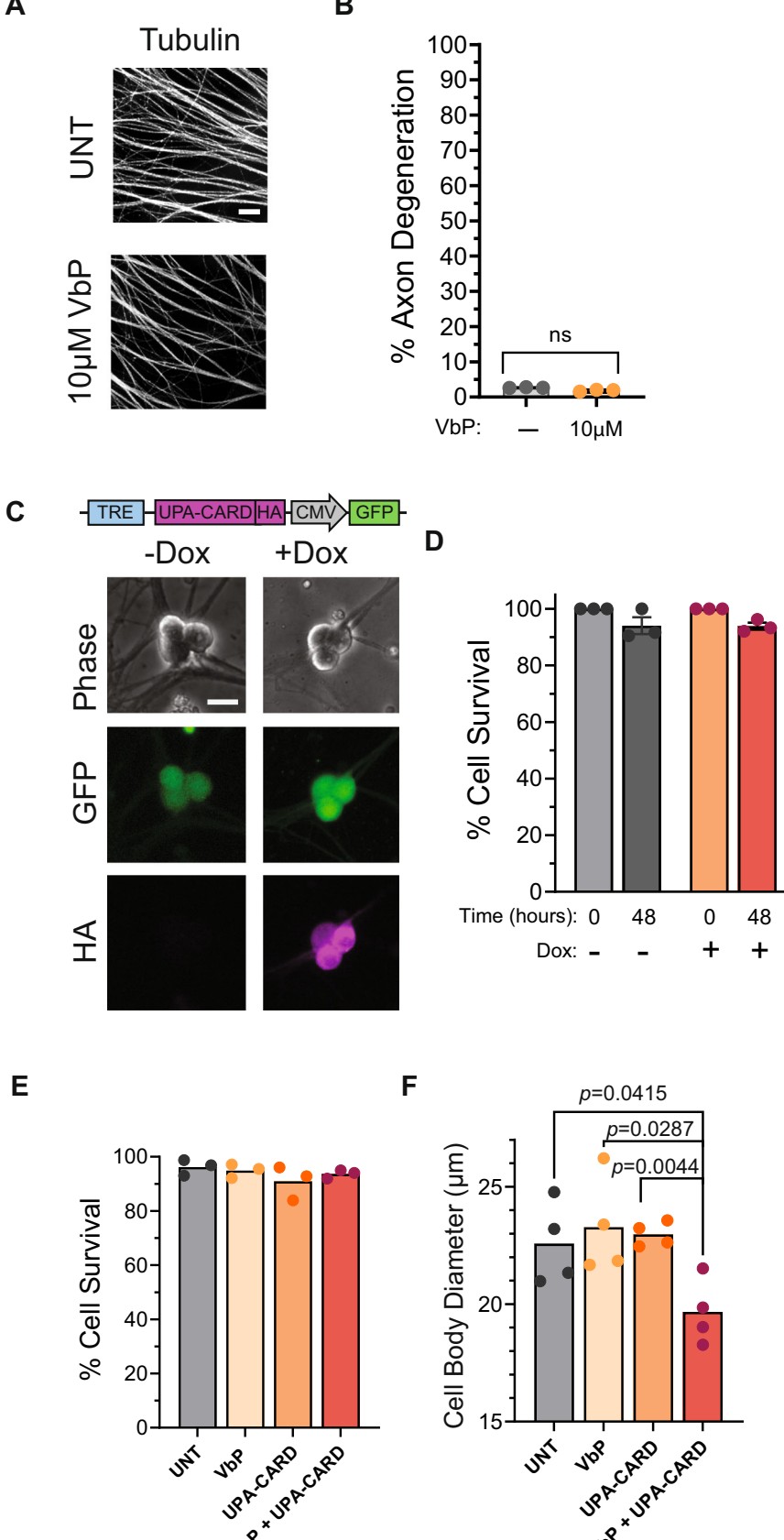

**Figure 4.   Neurons treated with both Val-boroPro and UPA-CARD overexpression display significantly smaller neuronal cell bodies.**

(A) Representative axonal images (stained with α-tubulin) in the axonal compartment of microfluidic chambers maintained in NGF or treated with 10 μM Val-boroPro (VbP) for 96 h. Scale bar = 20 μm. (B) Quantification of axon degeneration of experiments in (A) (n = 3 biological replicates). Individual data points and mean with standard error of mean represented. Statistical differences were examined by unpaired Student's t test. (C) Top: Diagram of the pLV–TRE–UPA CARD–CMV– EGFP plasmid, a doxycycline inducible system utilized to overexpress the NLRP1b UPA-CARD fragment in neurons. EGFP expression is controlled by the CMV promoter. Bottom: Representative phase and immunofluorescence images of neurons transduced with pLV–TRE–UPA CARD–CMV–EGFP and pLV-tTS/rtTA lentivirus, in the absence and presence of doxycycline (Dox; 10 μg/mL) for 72 h. Presence of the NLRP1 UPA-CARD is detected via the conjugated 3X HA tag and anti-HA immunostaining. Scale bar = 20 μm. (D) Quantification of cell survival of the experiments in (C) (n = 3 biological replicates). Individual data points and mean with standard error of mean represented. Statistical differences were examined by unpaired Student's t test. (E) Quantification of cell survival of neurons after 120 h of treatment with either VbP, UPA-CARD overexpression, or both together (n = 3 biological replicates). (F) Quantification of neuronal cell body diameter after 120 h of treatment with either VbP, UPA-CARD overexpression, or both together (n = 4 biological replicates). Individual data points and mean with standard error of mean represented. Statistical differences were examined by unpaired Student's t-test. Source data are available online for this figure.

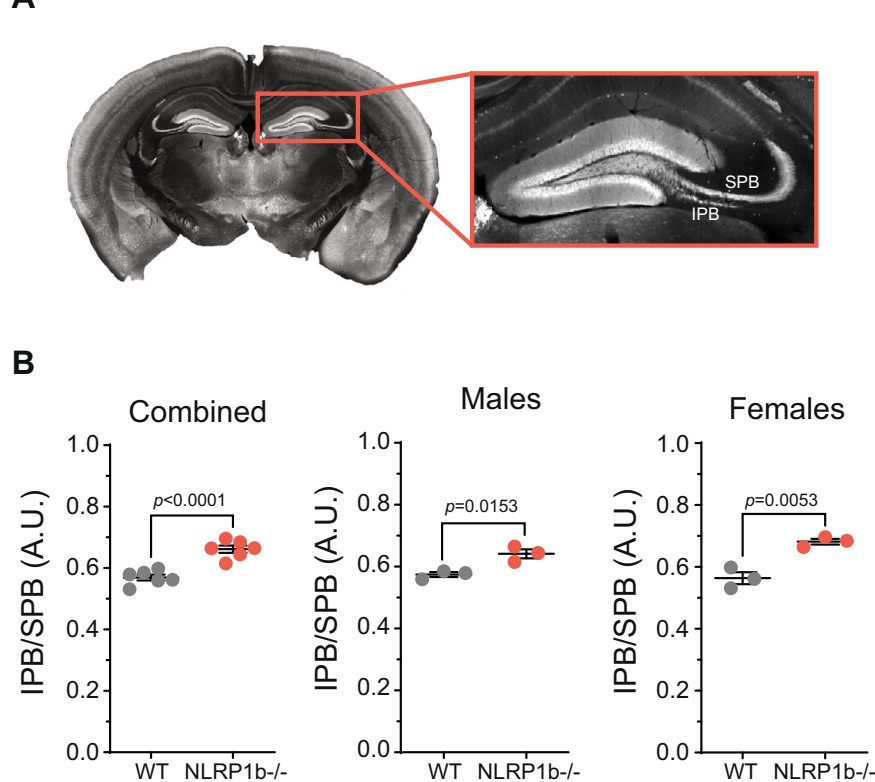

**Figure 5.   NLRP1 deficiency results in defects in axon pruning in vivo.**

(A) Representative image of coronal brain section of P45 wild-type mouse brain immunostained for Calbindin. Expanded image of the mouse hippocampus specifically highlighting the suprapyramidal bundle (SPB) and infrapyramidal bundle (IPB). Scale bar = 100 μm. (B) Quantification of IPB length relative to SPB length in wild-type and NLRP1b-deficient mice (Combined n = 6, Males n = 3, and Females n = 3 biological replicates). Individual data points represent a single animal with IPB length measured and averaged from 9 serial sections. Error bars represent standard error of mean. Source data are available online for this figure.

key role in AD progression. NLRP1 is particularly interesting as it is primarily expressed by neurons and is activated by amyloid β, leading to neuronal cell loss (Kaushal et al, 2015; Kummer et al, 2007; Tan et al, 2014). NLRP1 has also been implicated in other CNS pathologies such as traumatic brain injury, subarachnoid hemorrhage, and ischemic brain injury (Johnson et al, 2023; Mi et al, 2022). Based on our results, we propose that aberrant NLRP1 activation could result in pathological axon pruning further exacerbating neuronal injury and disease.

The mechanisms of NLRP1 activation have been reported in recent studies focusing primarily on immune cells. These studies have revealed that NLRP1 is activated by the release of a C-terminal fragment, known as the UPA-CARD, upon NLRP1 degradation *via* the proteasome (Lacey and Miao, 2019; Sandstrom et al, 2019). This mechanism of activation is a common theme of several pathogens (i.e. Anthrax, Shigella, Coronavirus,) (Barnett et al, 2023; Hartenian and Broz, 2022; Mitchell et al, 2019; Nozaki et al, 2022a; Sandstrom et al, 2019; Taabazuing et al, 2020) and stimuli (i.e., Reduced cytosolic ATP,

O$_3$, protein folding stress, reductive stress, dsRNA) (Orth-He et al, 2023; Robinson et al, 2022; Sandstrom, 2023; Zhou et al, 2023b) all known to activate NLRP1. NLRP1 can also be activated by chemical inhibitors (i.e., Val-boroPro) which disrupt the ability of dipeptidyl peptidase 8 and 9 (DPP8/9) to suppress NLRP1, resulting in spontaneous NLRP1 activation (Gai et al, 2019).

We have examined if NLRP1 activation during axon pruning occurs by similar mechanisms. Previously, we reported that proteasome inhibition with bortezomib or MG132 *enhances* axon pruning likely because of Caspase-6 (Casp6) stabilization (Cusack et al, 2013). Additionally, neither NLRP1 activation with VbP, nor UPA-CARD overexpression, was sufficient to induce axon degeneration in neurons. Similar to our results in neurons, VbP addition was also insufficient to activate NLRP1b in intestinal organoids (Mazzone et al, 2024). Thus, while NLRP1 is required for axon pruning, its activation alone is insufficient to cause axon degeneration or cell death in neurons. These results are not unexpected, as neurons have developed multiple mechanisms to restrict catastrophic degeneration pathways. This has been the best characterized in the context of neuronal apoptosis where it is established that neurons have multiple redundant mechanisms that keep apoptosis in check, and ensure the survival of the neuron (Hollville et al, 2019). Thus, it is not surprising that neurons also engage redundant mechanisms to keep axon pruning, and NLRP1, restricted.

Interestingly, combining Val-boroPro with UPA-CARD over-expression similarly did not induce cell death, though did result in significantly smaller neuronal cell bodies. This observed atrophy of the cell body is consistent with various studies which have reported a similar decrease in cell body diameter after axonal injury (i.e., axotomy) (Aydın et al, 2023; Li et al, 1998; Ota et al, 2002). While neuronal cell body shrinkage can be been attributed to a decrease in protein synthesis or cell stress (Deckwerth and Johnson, 1993; Deshmukh et al, 1996), recent work suggests this shrinkage could also be a survival mechanism (Aydın et al, 2023). These results emphasize the tight and contextual control of axon pruning neurons engage to prevent aberrant axon degeneration or cell death.

In immune cells NLRP1 is known to be activated by various bacteria and viruses which are absent in our model of axon pruning. As such, it is unknown exactly what stimuli NLRP1 senses during axon pruning to mediate degeneration. Some insight into how NLRP1 is activated during pruning can be gleaned from examining where in the axon pruning pathway NLRP1/Casp1 functions. We have found NLRP1 to function downstream of (or in parallel to) c-Jun phosphorylation (Fig. 3). Previously, we have also shown that inhibiting caspases with the broad-spectrum caspase inhibitor QVD-OPH (which also inhibits Casp1) during axon pruning still allows for the release of axonal cyt *c* (Cusack et al, 2013). These results suggest that Casp1, and thus also NLRP1, function downstream of mitochondrial outer membrane permeabilization (MOMP). We hypothesize that in the context of axon pruning, the release of mitochondrial proteins during MOMP, or MOMP mediated localized cellular stress, may serve as the triggers for NLRP1 activation.

Our work provokes an interesting question: why might an immune system sensor protein like NLRP1 be required for physiological axon pruning? We speculate that because NLRP1 is activated by various pathogens (i.e., viruses, bacteria), NLRP1/Casp1 in neurons might have evolved to detect neuro-invasive pathogens and degenerate targeted axons at the site of infection. This would prevent the retrograde trafficking of pathogen from axons to the cell body and reduce nervous system infection (Taylor and Enquist, 2015).

Consistent with this, a recent study has shown that deficiency in the axon pruning protein Casp6 resulted in less severe disease progression in models of skin HSV-1 infection (Li et al, 2022). Thus, NLPR1 may function both as a sentinel against pathogen infection, as well as regulate physiological neuronal plasticity and remodeling.

## Methods

**Reagents and tools table**

| Mouse models | Source | Catalog number |
|---|---|---|
| CASP1-/- (Casp11-/-) | Jackson Laboratory | 016621 |
| CASP1 -/- | Jackson Laboratory | 032662 |
| NLRP1b -/- | Jackson Laboratory | 021301 |
| NLRP3 -/- | Jackson Laboratory | 021302 |
| AIM2 -/- | Jackson Laboratory | 013144 |
| ASC -/- | Dr. V. Dixit, Genentech | MTA |
| CD-1/ICR | Charles River | 022 |

| qRT-PCR primers | Source | Sequence |
|---|---|---|
| GAPDH Fwd | Sigma | TGTGTCCGTCGTGGATCTGA |
| GAPDH Rev | Sigma | CCTGCTTCACCACCTTCTTGA |
| NLRP3 Fwd | Sigma | AGAAGAGACCACGGCAGAAG |
| NLRP3 Rev | Sigma | CCTTGGACCAGGTTCAGTGT |
| AIM2 Fwd | Sigma | AGGCTGCTACAGAAGTCTGTCC |
| AIM2 Rev | Sigma | TCAGCACCGTGACAACAAGTGG |
| ASC Fwd | Sigma | CTGCTCAGAGTACAGCCAGAAC |
| ASC Rev | Sigma | CTGTCCTTCAGTCAGCACACTG |
| CASP1 Fwd | Sigma | GGCACATTTCCAGGACTGACTG |
| CASP1 Rev | Sigma | GCAAGACGTGTACGAGTGGTTG |

| Recombinant DNA | | Source | Catalog number |
|---|---|---|---|
| pLV-TRE-mNeonGreen | | Vector Builder | NA |
| pLV-TRE-UPA-CARD-3xHA-CMV-GFP | | Vector Builder | NA |

| Software | Source | | Catalog number |
|---|---|---|---|
| Fiji (ImageJ) | https://imagej.net/software/fiji/ | | - |
| GraphPad Prism 10.4 | https://www.graphpad.com/ | | - |

| Antibodies | Source | Catalog number | Dilution |
|---|---|---|---|
| α-Tubulin | Sigma | T9026 | 1:500 |
| NeuN | Millipore | MAB377 | 1:500 |
| Calbindin D-28K | SWANT | CB38 | 1:1000 |
| Hoechst | Thermofisher | H3569 | 1:10,000 (IF)/ 1:5000 (IHC) |
| Phospho-c-Jun | Cell Signaling | 9261 | 1:250 |
| GFP | Abcam | ab13970 | 1:1000 |
| HA | Cell Signaling | 3724 | 1:1000 |

| Microfluidic components | Source | Catalog number | Final concentration |
|---|---|---|---|
| KRAYDEN SYLGARD 184 | Fisher | NC9285739 | – |

| Microfluidic components | Source | Catalog number | Final concentration |
|---|---|---|---|
| Poly-D-lysine (PDL) | Sigma | P7886 | 100 μg/mL |
| Laminin | Invitrogen | 23017-015 | 1 μg/mL |
| 0.5 M Borate Buffer | Boston Bioproducts | LT-66 | 0.1 M pH8.5 |
| 1.5 Glass Coverslips 25 ×25 mm | Corning | CLS285025 | – |

| Cell culture components | Source | Catalog number | Final concentration |
|---|---|---|---|
| Neurobasal Plus Culture System | Gibco, | A3653401 | 1× |
| Nerve Growth Factor (NGF) | Envigo | B.5017 | NBS-100:100 ng/mL AM-50:50 ng/mL |
| Fetal Bovine Serum (FBS) | Sigma | F2442 | NBS-100:10% AM-50:5% |
| Primocin | Invivogen | NV9141851 | NBS-100:1:500 |
| Aphidicolin | AG Scientific | A-1026 | 3.3 μg/mL |
| MEM | Gibco | 11095-080 | 1× |
| Penicillin/Streptomycin | Gibco | 600-5145 | AM-50/AM-0:100 μg/mL |
| FUDR | Sigma | F-0503 | AM-50/AM-0:0.02 mM |
| Caspase-1 Inhibitor Ac-YVAD-cmk | Sigma | SML0429-1MG | 5 μM, 10 μM |

## Mice

All animals are housed in appropriate conditions as approved by the UNC-Chapel Hill Department of Comparative Medicine (DCM). All animal experiments were approved and conducted in compliance with the University of North Carolina at Chapel Hill Institutional Animal Care and Use Committee (IACUC) under approval number 24-116. *Casp1/11−/−* (016621), *Casp1−/−* (032662), *NLRP1b−/−* (021301), *NLRP3-/-* (021302), and *AIM2−/−* (013144) mice are in the C57BL/6J genetic background and were obtained from The Jackson Laboratory. *ASC−/−* mice were kindly provided by Dr. V. Dixit (Genentech, South San Francisco, CA, USA). CD-1/ICR mice were ordered from Charles River and are routinely maintained in our lab. All knock-out mouse experiments were completed using appropriate wild-type controls.

## Primary sympathetic neuronal cultures

Primary mouse neurons were isolated from the superior cervical ganglion (SCG) of post-natal day 0-1 (P0-1) mice then plated into microfluidic chamber devices and cultured as previously described (Cusack et al, 2013). Specifically, SCG neurons were plated at a density of 10,000 cells per microfluidic chamber and cultured for 5 days in vitro (DIV) in Neurobasal Plus serum-containing chamber growth media (NBS-100). NBS-100: Neurobasal Plus Neuronal Culture System, 1:50 B27+, 100 ng/mL NGF, 10% fetal bovine serum, 0.02 mM FUDR, and 1:500 Primocin. Cultures were maintained with 3.3 μg/mL Aphidicolin (Nigrospora oryzae) for the first 3 DIV to eliminate non-neuronal cells.

## Culture and treatment of primary neurons in microfluidic devices

Primary sympathetic neurons were cultured in microfluidic devices as previously described (Cusack et al, 2013; Taylor et al, 2005) and maintained in NSB-100 until 5 DIV. Prior to treatment, axon and soma compartments were washed three times with either NGF containing media (AM-50), or NGF deficient media (AM-0). AM-50: MEM media, 5% fetal bovine serum, 50 ng/mL NGF, 100 μg/mL penicillin and 100 μg/mL streptomycin, and 0.02 mM FUDR. AM-0: MEM media, 5% fetal bovine serum, 100 μg/mL penicillin and 100 ug/mL streptomycin, and 0.02 mM FUDR. To induce apoptosis, both axon and soma compartments were maintained in AM-0 for 48 h. To induce axon pruning, the soma compartment was maintained in AM-50 while the axon compartment was maintained in AM-0 for 96 h. Importantly, for axon pruning experiments a 1 μL volume differential was established with the volume in the soma compartment being greater. This differential was reestablished every 12–20 hours to maintain fluidic isolation of the two compartments. Caspase-1 inhibitor, Ac-YVAD-cmk, was added to appropriate media at a final concentration of 5 μM and 10 μM. Val-boroPro (VbP) was added at a final concentration of 10 μM to both the soma and axon compartment. All experiments were performed with three biological replicates.

## Overexpression of active NLRP1 (UPA-CARD)

The doxycycline inducible NLRP1 overexpression plasmid was constructed by VectorBuilder using the pLV backbone. The expression of the NLRP1 UPA-CARD fragment (Sandstrom et al, 2019) conjugated to a 3× HA tag is controlled by a tetracycline responsive promoter (TRE), while EGFP is controlled by a CMV promoter. Tet regulatory elements tTS and rtTA were expressed in a separate plasmid constructed by VectorBuilder with BFP2 under the CMV promoter and tTS/rtTA under the mPGK promoter. Lentivirus was produced in HEK LTV-100 cells in AM-50 lacking FUDR. Neurons were treated with both the tTS/rtTA and the TRE-UPA-CARD viruses at a 1:1 ratio on P3 for 24 h with 6 μg/mL polybrene. Forty-eight hours after transduction, neurons were treated with 10 μg/mL doxycycline. At time of Dox induction all plates were imaged to assess the amount of live cells based on morphology at $t = 0$. Seventy-two hours after Dox induction neurons were collected, immunostained for HA, and assessed for cell survival (morphology).

## Quantification of cell survival and soma diameter after VbP and UPA-CARD treatment

Neurons were isolated from P0 CD-1 mice and plated into collagen coated ibidi 8-well chambers slides, or a 24-well plate. Neurons were transduced with both tTS/rtTA and the TRE-UPA-CARD viruses at a 1:1 ratio on P3 for 24 h with 6 μg/mL polybrene. Twenty-four hours after transduction, neurons were treated with 10 μg/mL doxycycline for 24 h at which time phase and GFP fluorescent images were taken for $t = 0$ (5 days in vitro). Cells were then treated with 10 μM VbP for a total of 120 h, after which phase and GFP images were again acquired. Neurons were then fixed with 3.8% formaldehyde and assessed for UPA-CARD expression based on HA immunostaining. Cell survival was quantified based on

morphology, comparing $t = 120$ to $t = 0$, finding the exact neuronal populations at each time point. Cell body diameter was quantified using $t = 120$ phase images (images collected before cells were fixed), by finding the major axis of each cell. Approximately 70–150 cells were quantified per condition and their averages calculated for four biological replicates. Statistical differences were calculated via GraphPad prism using an unpaired Student's $t$ test.

## Immunofluorescence in microfluidic chambers

Immunofluorescence staining was conducted in chambers as previously described (Cusack et al, 2013). All solutions for sample fixation and staining were added only to the top axon and soma reservoirs and allowed to flow into the chamber areas. A full list of the antibodies used for immunofluorescence can be found in the Reagents and Tools table.

## Image acquisition

Images were acquired on either a DMI6000 inverted fluorescent microscope (Leica) using an ORCA-ER B/W CCD camera (Hamamatsu; native resolution $1344 \times 1024$ pixels) through Metamorph software (Molecular Devices, version 7.6), or on a DMI8 inverted fluorescent microscope (Leica; native resolution $2048 \times 2048$ pixels) using a Leica DFC9000 sCMOS camera through LASX software (Leica).

## Axon degeneration quantification

Axon degeneration was quantified using immunofluorescent images of axons in microfluidic devices stained for α-tubulin. These images are analyzed using a published protocol to measure axon continuity/fragmentation (Sasaki et al, 2009). Briefly, multiple (4–5) images are acquired per microfluidic device to span the axon compartment. Images are thresholded and evaluated using the Analyze Particles plugin in ImageJ (FIJI) to generate a percent axon degeneration.

## Quantitative real time PCR

RNA isolation was performed using Arcturus PicoPure RNA Isolation Kit following manufacturer's protocol. cDNA was prepared using High Capacity cDNA Reverse transcription Kit using the kit protocol. Gene expression analysis was done by quantitative Real Time PCR using Power-up Sybr green chemistry.

## Infrapyramidal bundle assay

NLRP1b−/− and wild-type C57BL/6 littermates (3 males and 3 females per genotype) were anesthetized with isofluorane and perfused with 4% formaldehyde at post-natal day 45 (P45). Whole brain tissue was then collected from these mice and sectioned to 60 μM thick sections using a Leica VT 1000S vibrating microtome. For each mouse, nine serially collected samples spanning the IPB were then stained using the following primary antibodies: NeuN, Calbindin D-28K, and Hoechst 33258. Brain sections were imaged on a Leica DMI8 inverted fluorescent microscope using a Leica DFC9000 sCMOS camera (native resolution 2048 ×2048 pixels) and acquired through LASX software (Leica) using a Leica HCX PL

FLUOTAR 10x/0.30na PH1 Objective. Image names were encrypted using the ImageJ plugin "Blind Analysis Tools", and Relative IPB length (A.U) for each mouse was obtained by averaging the relative IPB length of the nine serial brain sliced. Relative IPB length is measured by taking the ratio of the length of the IPB to the length of the suprapyramidal bundle (SPB).

## Statistics

All experiments were completed with 3–4 biological replicates as indicated in the figure legends. For in vitro experiments, statistical significance was calculated using an unpaired two-tailed $T$ test comparing treatment to untreated control, or deficient neurons compared to wild-type controls. For in vivo experiments, statistical significance was calculated using an unpaired two-tailed $T$ test comparing NLRP1-deficient brains to wild-type littermate controls.

# Data availability

This study includes no data deposited in external repositories.

The source data of this paper are collected in the following database record: biostudies:S-SCDT-10_1038-S44319-025-00402-y.

# Peer review information

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

## Acknowledgements

We thank the Deshmukh lab members for their discussions of this work and critical review of this manuscript. Funding: This work was supported by the National Institute of Neurological Disorders and Stroke (NINDS/NIH) grant NS117133.

## Author contributions

**Selena E Romero**: Conceptualization; Data curation; Formal analysis; Supervision; Validation; Investigation; Visualization; Methodology; Writing—original draft; Project administration; Writing—review and editing. **Matthew J Geden**: Data curation; Formal analysis; Visualization; Writing—review and editing. **Richa Basundra**: Data curation; Formal analysis; Visualization. **Kiran Kelly-Rajan**: Data curation. **Edward A Miao**: Conceptualization; Resources; Writing—review and editing. **Mohanish Deshmukh**: Conceptualization; Supervision; Funding acquisition; Writing—original draft; Project administration; Writing—review and editing.

Source data underlying figure panels in this paper may have individual authorship assigned. Where available, figure panel/source data authorship is listed in the following database record: biostudies:S-SCDT-10_1038-S44319-025-00402-y.

## Disclosure and competing interests statement

The authors declare no competing interests.

# Expanded View Figure

**A**

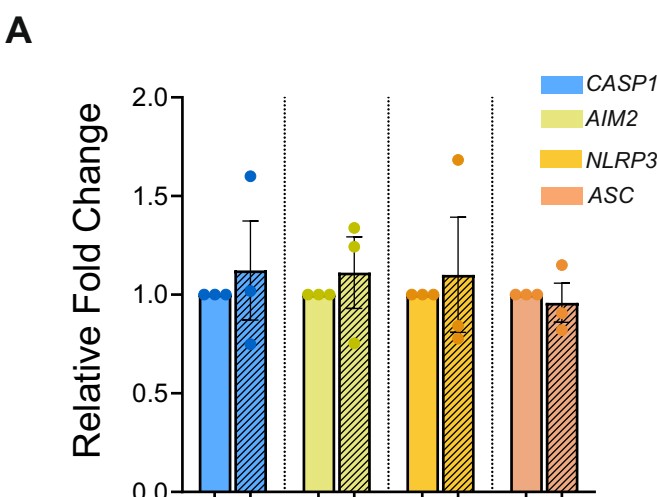

**Figure EV1. Inflammasome components remain intact in the NLRP1-deficient sympathetic neurons.**

(A) Fold change expression of *CASP1*, *AIM2*, *NLRP3*, and *ASC* in wild-type and NLRP1b-deficient sympathetic neurons, normalized to wild-type. Individual data points and mean with standard error of mean represented. Statistical differences were examined by unpaired Student's *t* test ($n = 3$ biological replicates). Source data are available online for this figure.

