## [Peer Review File · EMBO Reports]

The NLRP1 inflammasome is an essential and selective mediator of axon pruning in neurons

Mohanish Deshmukh, Selena Romero, Matthew Geden, Richa Basundra, Kiran Kelly-Rajan, and Edward Miao

Corresponding author(s): Mohanish Deshmukh (mohanish@med.unc.edu)

Review Timeline:

Submission Date:	13th Oct 23
Editorial Decision:	11th Dec 23
Revision Received:	27th Dec 24
Editorial Decision:	24th Jan 25
Revision Received:	28th Jan 25
Accepted:	31st Jan 25

Editor: *Martina Rembold*

Transaction Report:

Dear Dr. Deshmukh

Thank you for the submission of your research manuscript to our journal. We have now received the full set of referee reports that is copied below.

As you will see, the referees acknowledge that the findings are potentially interesting, but they also raise a number of concerns that need to be addressed.

While we do not mandate mechanistic insight at EMBO Reports, the suggestions from referee 2 to qualify the involvement of Jun, test whether NLRP1 is required for caspase activation and whether components of the NGF pathway are involved should be addressed. Please also note that per our editorial policies we do not allow "data not shown". If you make conclusions based on data at hand, then please include the data in the manuscript.

Rescue experiments as suggested by referee 1 and a specificity test of the Casp1 inhibitor (referee 3) should be performed. Also all other concerns from referee 3 need to be addressed.

Given these constructive comments, we would like to invite you to revise your manuscript with the understanding that the referee concerns (as detailed above and in their reports) must be fully addressed and their suggestions taken on board. Please address all referee concerns in a complete point-by-point response. Acceptance of the manuscript will depend on a positive outcome of a second round of review. It is EMBO Reports policy to allow a single round of revision only and acceptance or rejection of the manuscript will therefore depend on the completeness of your responses included in the next, final version of the manuscript.

We realize that it is difficult to revise to a specific deadline. In the interest of protecting the conceptual advance provided by the work, we recommend a revision within 3 months (March 11). Please discuss the revision progress ahead of this time with the editor if you require more time to complete the revisions.

I am also happy to discuss the revision further via e-mail or a video call, if you wish.

*******IMPORTANT NOTE:**

We perform an initial quality control of all revised manuscripts before re-review. Your manuscript will FAIL this control and the handling will be delayed IN CASE the following APPLIES:

- 1) A data availability section providing access to data deposited in public databases is missing. If you have not deposited any data, please add a sentence to the data availability section that explains that.
- 2) Your manuscript contains statistics and error bars based on $n=2$. Please use scatter blots in these cases. No statistics should be calculated if $n=2$.

When submitting your revised manuscript, please carefully review the instructions that follow below. Failure to include requested items will delay the evaluation of your revision. *****

2) individual production quality figure files as .eps, .tif, .jpg (one file per figure).

Please download our Figure Preparation Guidelines (figure preparation pdf) from our Author Guidelines pages <https://www.embopress.org/page/journal/14693178/authorguide> for more info on how to prepare your figures.

4) a complete author checklist, which you can download from our author guidelines (). Please insert information in the checklist that is also reflected in the manuscript. The completed author checklist will also be part of the RPF.

5) Please note that all corresponding authors are required to supply an ORCID ID for their name upon submission of a revised manuscript (). Please find instructions on how to link your ORCID ID to your account in our manuscript tracking system in our Author guidelines ()

6) We replaced Supplementary Information with Expanded View (EV) Figures and Tables that are collapsible/expandable online. A maximum of 5 EV Figures can be typeset. EV Figures should be cited as 'Figure EV1, Figure EV2' etc... in the text and their respective legends should be included in the main text after the legends of regular figures.

7) Please note that a Data Availability section at the end of Materials and Methods is now mandatory. In case you have no data that requires deposition in a public database, please state so instead of refereeing to the database. See also < <https://www.embopress.org/page/journal/14693178/authorguide#dataavailability>>. Please note that the Data Availability Section is restricted to new primary data that are part of this study.

Additional information on source data and instruction on how to label the files are available .

10) Figure legends and data quantification:
The following points must be specified in each figure legend:

- the name of the statistical test used to generate error bars and P values,
- the number (n) of independent experiments (please specify technical or biological replicates) underlying each data point,
- the nature of the bars and error bars (s.d., s.e.m.)
- If the data are obtained from n {less than or equal to} 5, show the individual data points in addition to the SD or SEM.
- If the data are obtained from n {less than or equal to} 2, use scatter blots showing the individual data points.

See also the guidelines for figure legend preparation:
<https://www.embopress.org/page/journal/14693178/authorguide#figureformat>

11) All Materials and Methods need to be described in the main text. We would encourage you to use 'Structured Methods', our new Materials and Methods format. According to this format, the Materials and Methods section should include a Reagents and Tools Table (listing key reagents, experimental models, software and relevant equipment and including their sources and relevant identifiers) followed by a Methods and Protocols section in which we encourage the authors to describe their methods using a step-by-step protocol format with bullet points, to facilitate the adoption of the methodologies across labs.

More information on how to adhere to this format as well as downloadable templates (.doc or .xls) for the Reagents and Tools Table can be found in our author guidelines: < <https://www.embopress.org/page/journal/14693178/authorguide#manuscriptpreparation>>. An example of a Method paper with

Structured Methods can be found here: .

12) Our journal encourages inclusion of *data citations in the reference list* to directly cite datasets that were re-used and obtained from public databases. Data citations in the article text are distinct from normal bibliographical citations and should directly link to the database records from which the data can be accessed. In the main text, data citations are formatted as follows: "Data ref: Smith et al, 2001" or "Data ref: NCBI Sequence Read Archive PRJNA342805, 2017". In the Reference list, data citations must be labeled with "[DATASET]". A data reference must provide the database name, accession number/identifiers and a resolvable link to the landing page from which the data can be accessed at the end of the reference. Further instructions are available at .

13) As part of the EMBO publication's Transparent Editorial Process, EMBO Reports publishes online a Review Process File to accompany accepted manuscripts. This File will be published in conjunction with your paper and will include the referee reports, your point-by-point response and all pertinent correspondence relating to the manuscript.

Kind regards,

Referee #1:

This manuscript by Romero describes the role of the NLRP1 inflammasome in regulating axon pruning. They first show that Caspase-1 is required for axon pruning but not for apoptosis in compartmentalized neuronal cultures. By using both Casp1 inhibitors and Casp1 KO neurons, they found that Casp1 plays an essential role in axon pruning. They further asked the question of what inflammasomes are required for casp1 activation. Among several potential sensors, they found that NLRP1 is required for axon pruning. However, ectopic activation of NLRP1 activation alone is not sufficient to induce axon pruning or neuronal death, suggesting that NLRP1 is a permissive factor. Finally, they show that NLRP1 also regulates axon pruning of the Infrapyramidal bundle in vivo. Overall, the study reports a possible role of Caspase-1 and the inflammasome NLRP1 in axon pruning. Although the findings appear to be interesting, the current study lacks some mechanistic insights into some important questions, how the inflammasome NLRP1 and Caspase1 is activated during remodeling? Where is caspase1 activated in neurons, exclusively in axons or in both axons and soma. Can the author design a bio-sensor to detect caspase-1 activity in pruning neurons? Several technical issues also exist, for example, they need to overexpress NLRP1 in the NLRP1-deficient neuron to rescue the defects in axon pruning.

Referee #2:

Romero et al investigate one of the classic questions in developmental neurobiology-the mechanism of developmental axon pruning using the NGF deprivation model. When sympathetic neurons are derived of NGF, they undergo apoptosis. When sympathetic axons are deprived of NGF (in a compartment chamber) they undergo axon degeneration. While there are many molecular similarities between these two processes including a role for caspases, the mechanism of caspase activation in axon pruning was previously unknown. Here Romero and colleagues solve this mystery. They look to innate immunity, where a variety of mechanisms have been described for non-apoptotic caspase activation. They test all the known candidates, and find that the NLRP1 inflammasome is the long-missing caspase activator in developmental axon pruning. Their data are crystal clear and the mutant neurons have a very strong axon pruning phenotype. There is also an in vivo developmental pruning phenotype. The data presented are clear and convincing and the paper is very well written. It is a short format paper with one clear and exciting finding, and so is a great fit for EMBO Reports. While I realize that mechanism is not necessary for EMBO reports, the

authors do describe some mechanistic work as "data not shown" in the discussion. I would urge to authors to add one more figure with a couple of simple mechanistic studies. For examples, the authors say that NLRP1 functions downstream of phosphorylated Jun. I imagine the authors are saying that Jun is still phosphorylated in the absence of NLRP1, however this does not mean that NLRP1 is downstream, it could be parallel. To be downstream, then NLRP1 activation would require pJun. Is this what the authors are saying?

I would include a figure with some simple mechanistic analysis of NLRP1. Most importantly, is NLRP1 required for caspase activation? The data for axon pruning is great, but it is disappointing that there is not confirmation that there is a concomitant defect in caspase activation. Second, is NLRP1 cleaved (canonical activation mechanism) following NGF deprivation? If so it would be easy to ask if NLRP1 activation requires known components of the NGF deprivation pathway (like DLK and JNK). The authors should also show whether or not Jun is still phosphorylated? A few fairly straightforward experiments like this would take what is a very nice paper and make it a great paper.

Referee #3:

The authors show that the inflammasome component NLRP1 is necessary but not sufficient for NGF deprivation-dependent axon degeneration in cultured sympathetic neurons and for infrapyramidal bundle pruning in mouse hippocampus. This occurs through NLRP1-dependent activation of Caspase 1 and is distinct from apoptotic death of neurons after NGF withdrawal. The strengths of this manuscript are the use of microfluidics to isolate soma vs. axonal effects and in vivo confirmation for role of NLRP1. Overall the manuscript is well written and results are appropriately described. Further showing a specific role for NLRP1 but not other inflammasome components that can activate Casp1 brings an advance for the field. However, several points detract from my enthusiasm for this manuscript as submitted and I think raise some questions for interpretation of the data as it stands.

1. The authors use inhibitors of Casp1 to conclude that Casp1 but not Casp11 or other caspases are needed for NGF-withdrawal dependent pruning of axons. No specificity data are shown for the inhibitor to confirm that Casp11 activity (or other caspases) is not impacted by the Ac-YVAD-cmk reagent. I think that this is a critical control in that the Casp1 knockout neurons also have deficient Casp11 as noted by the authors.
2. The authors use Aim2^{-/-}, Nlrp3^{-/-} and ASC^{-/-} sympathetic neurons to conclude that only Nlrp1 is needed for Casp1 activation in these neurons after NGF withdrawal. This conclusion would be strengthened by data showing that Aim2, Nlrp3 and ASC are expressed in the sympathetic neurons used here and that these proteins as well as Nlrp1 localize to the axonal compartment where Casp1 is presumably activated. Critical here is whether Nlrp1 is needed because the other components are not expressed in the sympathetic neurons used.
3. The authors specifically state in the results that Nlrp1b deficient mice are on the 129 mouse strain which lacks Nlrp1a and Nlrp1c so they are devoid of NLRP1 activity. Yet the methods indicate that all mouse strains used were on a C57Bl/6 background. This needs to be clarified and the data would be strengthened by showing that other Casp1 activating pathways (Aim2, Nlrp3, and ASC) remain intact in the sympathetic neurons from these animals.
4. The studies in Fig 3 need to show that VbP treatment and UPA-CARD overexpression are in fact doing what the authors anticipate in their experimental system. Assuming the authors interpretation is correct, I think they should speculate on why NLRP1 activation alone is not sufficient to induce axon degeneration in the discussion.

Referee #1:

This manuscript by Romero describes the role of the NLRP1 inflammasome in regulating axon pruning. They first show that Caspase-1 is required for axon pruning but not for apoptosis in compartmentalized neuronal cultures. By using both Casp1 inhibitors and Casp1 KO neurons, they found that Casp1 plays an essential role in axon pruning. They further asked the question of what inflammasomes are required for casp1 activation. Among several potential sensors, they found that NLRP1 is required for axon pruning. However, ectopic activation of NLRP1 activation alone is not sufficient to induce axon pruning or neuronal death, suggesting that NLRP1 is a permissive factor. Finally, they show that NLRP1 also regulates axon pruning of the Infrapyramidal bundle in vivo. Overall, the study reports a possible role of Caspase-1 and the inflammasome NLRP1 in axon pruning. Although the findings appear to be interesting, the current study lacks some mechanistic insights into some important questions, how the inflammasome NLRP1 and Caspase1 is activated during remodeling? Where is caspase1 activated in neurons, exclusively in axons or in both axons and soma. Can the author design a bio-sensor to detect caspase-1 activity in pruning neurons? Several technical issues also exist, for example, they need to overexpress NLRP1 in the NLRP1-deficient neuron to rescue the defects in axon pruning.

1. Mechanism of inflammasome activation in the context of pruning.

We agree that this is an important question which we too are very interested in examining, but one for which we currently do not have any insight. The most unbiased approach would be to conduct biochemical experiments to identify NLRP1-interacting proteins specifically during pruning. However, this is difficult to conduct in the microfluidic chambers which contain limiting material. As described in the manuscript, we have attempted to examine whether the mechanism by which NLRP1 is activated during axon pruning is similar to that observed in the context of pathogen infection. These experiments include examining the outcome of proteasome inhibition (since NLRP1 can be activated *via* its partial degradation in the proteasome) and inhibiting DPP8/9 with Val-boroPro. However, these experiments have not provided clear answers to this question. Our plans are to scale up the model of axon pruning with the redesign of microfluidic chambers or using Campenot chambers and attempt the biochemical approach. We hope the reviewer agrees that these long-term experiments are appropriate for a follow-up publication.

2. Caspase-1 activity during axon pruning.

We have attempted to examine this *via* multiple approaches. For example, we tested various Caspase-1 antibodies to examine where Caspase-1 is localized during axon pruning. However, either these antibodies gave no signal in these neurons or gave a false positive signal that was still present in the Caspase-1 KO neurons. We also attempted to examine Caspase-1 activity with the FAM-FLICA reagent. While we obtained a signal in the soma during axon pruning, here too, the signal was present in the Caspase-1 KO neurons. So, unfortunately, we were unable to evaluate this question with rigor. As described in our response to Reviewer 3, we have now examined the specific importance of Caspase-1 in axon pruning and report that mice deleted for Casapse-1 alone are unable to undergo axon pruning (new Fig. 1D, E).

3. Re-expression of NLRP1 in the NLRP1 KO neurons.

We acknowledge that rescuing the NLRP1 KO phenotype by over-expression of WT NLRP1 would strengthen our studies. We spent 2.5 years attempting to do exactly this with no success due to technical limitations. Sympathetic neurons are nearly impossible to transfect, with only ~5% of neurons successfully transfected using multiple mechanisms (*i.e.* Neon electroporation, Lipofectamine 2000/3000/LTX, etc.). This has left viral transduction either by AAV or Lentivirus as our only remaining avenue. Unfortunately, NLRP1 is relatively large (~140 kDa) and is unsuitable for AAV production. Rather frustratingly, we also experienced difficulties in making lentivirus which expresses WT NLRP1, likely because the large size of the viral plasmid (~12 kb) which does not effectively package and make viral particles. Indeed, we created 4 different viral plasmids in our lab and had a commercial vendor create an additional 3 lentiviral plasmids for NLRP1 expression. We attempted to generate viral particles with the established techniques in our lab, a commercial vendor, and a Lentiviral Core facility at UNC. However, none of these attempts produced a virus that was able to transduce neurons (control virus however worked beautifully). Thus, while reintroduction of NLRP1 into the NLRP1 KO neurons should have been a straightforward question to address, it has been technically challenging despite our significant efforts.

We recognize that we have been unable to address any of the Reviewer 1 comments. We hope this Reviewer acknowledges the appropriateness of our manuscript not only for the novelty of the results but also for some of the additional experiments that we have been able to conduct (described below) in this revised manuscript.

Referee #2:

Romero et al investigate one of the classic questions in developmental neurobiology-the mechanism of developmental axon pruning using the NGF deprivation model. When sympathetic neurons are deprived of NGF, they undergo apoptosis. When sympathetic axons are deprived of NGF (in a compartment chamber) they undergo axon degeneration. While there are many molecular similarities between these two processes including a role for caspases, the mechanism of caspase activation in axon pruning was previously unknown. Here Romero and colleagues solve this mystery. They look to innate immunity, where a variety of mechanisms have been described for non-apoptotic caspase activation. They test all the known candidates, and find that the NLRP1 inflammasome is the long-missing caspase activator in developmental axon pruning. Their data are crystal clear and the mutant neurons have a very strong axon pruning phenotype. There is also an in vivo developmental pruning phenotype.

The data presented are clear and convincing and the paper is very well written. It is a short format paper with one clear and exciting finding, and so is a great fit for EMBO Reports. While I realize that mechanism is not necessary for EMBO reports, the authors do describe some mechanistic work as "data not shown" in the discussion. I would urge to authors to add one more figure with a couple of simple mechanistic studies. For examples, the authors say that NLRP1 functions downstream of phosphorylated Jun. I imagine the authors are saying that Jun is still phosphorylated in the absence of NLRP1, however this does not mean that NLRP1 is downstream, it could be parallel. To be downstream, then NLRP1 activation would require pJun. Is this what the authors are saying?

I would include a figure with some simple mechanistic analysis of NLRP1. Most importantly, is NLRP1 required for caspase activation? The data for axon pruning is great, but it is disappointing that there is not confirmation that there is a concomitant defect in caspase activation. Second, is NLRP1 cleaved (canonical activation mechanism) following NGF deprivation? If so it would be easy to ask if NLRP1 activation requires known components of the NGF deprivation pathway (like DLK and JNK). The authors should also show whether or not Jun is still phosphorylated? A few fairly straightforward experiments like this would take what is a very nice paper and make it a great paper.

1. Status of c-Jun phosphorylation and caspase activation in the NLRP1 KO neurons during axon pruning.

We thank the reviewer for their positive comments and this suggestion. We have now included the c-Jun phosphorylation data (new Fig. 3) that was previously not shown. These results show similar levels of c-Jun phosphorylation in the WT and NLRP1 KO neurons during axon pruning. As the reviewer correctly indicates, these results show that NLRP1 functions either downstream of, or in parallel to, c-jun phosphorylation during axon pruning.

We also conducted experiments to examine if caspases are activated in the NLRP1 KO neurons during axon pruning. In our previous publication (Cusack et al. 2013), we had used the cleaved caspase-3 (Cell Signaling) to detect caspase activation during axon pruning. While the current version of this antibody from Cell Signaling still detects caspase-3 activation in the nucleus in the context of apoptosis, we were unfortunately unable to detect reliable signal in the axons even in WT neurons. Likewise, we attempted to probe for cleaved caspase-6 using various antibodies (the cleaved caspase-6 antibodies from Santa Cruz used in our previous publication are no longer available), but here too we could not obtain a reliable, specific signal for cleaved caspase-6 in the axons undergoing pruning. Thus, we were unfortunately unable to assess the status of caspase activation in the NLRP1 KO axons during pruning.

2. Assessing NLRP1 cleavage during axon pruning.

Previous studies that have examined NLRP1 cleavage in immune cells in the context of pathogen infection have conducted Western blots to discern specific cleavage products after NLRP1 activation. As we described earlier, we are not able to obtain enough material from neurons with our current design of the microfluidic chambers. We attempted to circumvent this problem by reintroducing a version of WT NLRP1 which has a N-terminal HA tag, and a C-terminal 3X Flag tag. If NLRP1 is indeed cleaved during axon pruning, we anticipated that the HA signal should decrease as it gets cleaved and degraded which the 3X Flag tag should persist. This approach was used by the Vance

Lab in transfected HEK293T cells. We obtained the same plasmid from the Vance lab, but, as indicated in our response to Review 1, we have had significant difficulties in expressing NLRP1 in neurons. This was frustrating for us because we had also generated NLRP1 mutant constructs (in the FIIND domain) that could not be activated by cleavage and proteasomal degradation, which we were also unable to evaluate in the axon pruning context.

Referee #3:

The authors show that the inflammasome component NLRP1 is necessary but not sufficient for NGF deprivation-dependent axon degeneration in cultured sympathetic neurons and for infrapyramidal bundle pruning in mouse hippocampus. This occurs through NLRP1-dependent activation of Caspase 1 and is distinct from apoptotic death of neurons after NGF withdrawal. The strengths of this manuscript are the use of microfluidics to isolate soma vs. axonal effects and in vivo confirmation for role of NLRP1. Overall the manuscript is well written and results are appropriately described. Further showing a specific role for NLRP1 but not other inflammasome components that can activate Casp1 brings an advance for the field. However, several points detract from my enthusiasm for this manuscript as submitted and I think raise some questions for interpretation of the data as it stands.

1. The authors use inhibitors of Casp1 to conclude that Casp1 but not Casp11 or other caspases are needed for NGF-withdrawal dependent pruning of axons. No specificity data are shown for the inhibitor to confirm that Casp11 activity (or other caspases) is not impacted by the Ac-YVAD-cmk reagent. I think that this is a critical control in that the Casp1 knockout neurons also have deficient Casp11 as noted by the authors.
We agree that it is important to definitively evaluate the specific importance of Caspase-1 (and not Caspase-11) in axon pruning. While previous publications have shown that Ac-YVAD-cmk is selective to Caspase-1 and not Caspase-11 (Kang et al., Cell Death Differ, 2002), we have now examined this question using the Caspase-1 selective KO mice in which Caspase-11 is still intact (Rauch et al., Immunity, 2017). Our new results show that Caspase-1 deficiency alone prevents axon pruning (new Fig. 1D, E)
2. The authors use Aim2^{-/-}, Nlrp3^{-/-} and ASC^{-/-} sympathetic neurons to conclude that only Nlrp1 is needed for Casp1 activation in these neurons after NGF withdrawal. This conclusion would be strengthened by data showing that Aim2, Nlrp3 and ASC are expressed in the sympathetic neurons used here and that these proteins as well as Nlrp1 localize to the axonal compartment where Casp1 is presumably activated. Critical here is whether Nlrp1 is needed because the other components are not expressed in the sympathetic neurons used.
We thank the reviewer for this suggestion. We have examined the expression of AIM2, NLRP3, ASC, and Caspase-1 in sympathetic neurons. Our results show that these inflammasome components are indeed expressed in these neurons (new Fig. 2B).
3. The authors specifically state in the results that Nlrp1b deficient mice are on the 129 mouse strain which lacks Nlrp1a and Nlrp1c so they are devoid of NLRP1 activity. Yet the methods indicate that all mouse strains used were on a C57Bl/6 background. This needs to be clarified and the data would be strengthened by showing that other Casp1 activating pathways (Aim2, Nlrp3, and ASC) remain intact in the sympathetic neurons from these animals.
We agree. As indicated, the mice we used in our experiments were deleted for NLRP1b in the 129 background that lack NLRP1a and NLRP1c activity (Kovarova et al., J. Immunol, 2012). The Jax website indicates that these mice were backcrossed into the C57/BL6 background and still maintain complete NLRP1 deficiency. However, since these mice were specifically deleted for NLRP1b, and to distinguish these mice from a newer generated mouse with complete deletion of NLRP1 (a, b, and c), we refer to the mice we use in our manuscript as NLRP1b KO mice for clarity.

We also agree that it is good to show that the other inflammasomes remain intact in the NLRP1b KO sympathetic neurons. We have now included a new figure which shows that AIM2, NLRP3, ASC, and Caspase-1 remain expressed in the NLRP1b KO neurons (new Fig. EV1).

4. The studies in Fig 3 need to show that VbP treatment and UPA-CARD overexpression are in fact doing what the authors anticipate in their experimental system. Assuming the authors interpretation is correct, I think they should speculate on why NLRP1 activation alone is not sufficient to induce axon degeneration in the discussion.

We agree that it would be good to show that VbP and UPA-CARD are functionally active in neurons even though they have no effect on neuronal survival. However, there is no straightforward way to evaluate the function of these in neurons. For example, VbP is an inhibitor of DPP8/DPP9, but does not lead to their degradation. In immune cells, the efficacy of VbP is demonstrated by assaying the release of LDH (an assay for cell death) with VbP treatment. For UPA-CARD overexpression, we demonstrate that this constitutively active NLRP1 is expressed specifically upon the addition of Doxycycline.

Importantly, in new experiments we have examined the outcome of VbP addition and UPA-CARD overexpression together in neurons. While this combined treatment was also not sufficient to induce neurodegeneration, it resulted in neuronal atrophy. We have quantified this result and have included these new results in Fig. 4F. As suggested, we have included a discussion of why NLRP1 activation alone may not be sufficient to induce degeneration in neurons in the revised manuscript.

Dear Dr. Deshmukh

Thank you for the submission of your revised manuscript to EMBO reports. We have now received the reports from the referees who were asked to assess it (copied below).

As you can see, both referees find that the study has been significantly improved during revision and recommend publication.

Before I can accept the manuscript, I need you to address some minor points below:

- Your manuscript will be published in our Reports section and therefore needs a combined "Results and Discussion" section (with this subheading).
 - Please provide up to 5 keywords.
 - Please update the 'Conflict of interest' paragraph to our new 'Disclosure and competing interests statement'. For more information see <https://www.embopress.org/page/journal/14693178/authorguide#conflictsofinterest>
 - Regarding the Author Contributions, we now use CRediT to specify the contributions of each author in the journal submission system. Therefore, please remove the Author Contributions from the manuscript file and make sure that the author contributions in our online manuscript tracking system are correct and up-to-date. The information you specified in the system will be automatically retrieved and typeset into the article. You can enter additional information in the free text box provided, if you wish.
 - Please remove all figures from the manuscript text file and upload them as individual production quality figure files (.eps, .tif, .jpg, one file per figure). The figure legends should stay in the manuscript file at the end, after the References. First the "Figure Legends" for the main figures, followed by the "Expanded View Figure Legends".
 - Please provide a callout for Figure 4F in the text where applicable.
 - You refer to "supplementary tables" on p 19. I assume this should refer to the Reagents and Tools table. Please correct the callout.
 - Figure 1D: A spot-check of the source data indicated that the NGF Maint. image you provided for CASP1 KO seems not to match to the one shown in the figure panel. It looks similar but not identical. Please check.
 - Please remove the Reagents and Tools table from the manuscript and upload it as separate file.
 - You mention the use of a select agent in the Author Checklist in the section "Dual Use Research of Concern". Could you please clarify to which agent this applies? Thank you.
 - Materials and Methods should be Methods
 - We perform a routine image integrity check on all revised manuscripts. Doing so, we noticed that the Wildtype images shown in Figure 2C have been reused in Figure 2E. Please only do so, if the samples were all from the same experiments and the WT staining is thus the appropriate control for all mutant conditions shown. If this is the case and you wish to keep the single WT example, please clearly state the re-use and the correct use as a control in the figure legend. An alternative WT image to show reproducibility is preferred. Please also update the Source Data in case you decide to replace the image.
 - Our production/data editors have asked you to clarify several points in the figure legends (see below). Please incorporate these changes in the manuscript and return the revised file with tracked changes with your final manuscript submission.
- A) Statistical test information. Only p-values that are actually shown in the figure panel(s) should (and must) be defined in the legends, all others should be removed from (or added to) the legend. Moreover, we ask for the specification of exact p-values: 1. Please note that the exact p values are not provided in the legends of figures 1C, E; 2D, F; 4F, 5B.
- B) Replicates and error bars:
2. Please note that information related to n is missing in the legend of figure 2B.
 3. Please note that the error bars are not defined in the legend of figure 2B.
 4. Please note that scale bar and its definition are missing for figure 3A.
 5. Please define the nature of the replicates (biological, technical) in all figure legends and ensure not to have done statistical analysis on technical replicates.

- Abstract: please do not insert references in the Abstract. If you remove them, please ensure that the relevant literature is cited in the Introduction.

- Finally, EMBO Reports papers are accompanied online by

A) a short (1-2 sentences) summary of the findings and their significance,

B) 2-3 bullet points highlighting key results and

C) a schematic summary figure that provides a sketch of the major findings (not a data image).

Please provide the summary figure as a separate file in PNG or JPG format at a size of 550x300-600 pixels (width x height).

Please note that the size is rather small and that text needs to be readable at the final size. Please send us this information along with the revised manuscript.

With kind regards,

Martina Rembold, PhD

Senior Editor

EMBO reports

=====

Referee #2:

The authors have significantly improved the manuscript with new mechanistic insights and addition of important controls, including analysis from a new KO mice. While it is unfortunate that they were not able to perform the rescue experiments requested by reviewer 1 (despite Herculean efforts), I believe that the weight of data is strong enough to make this a compelling and complete study that will be of interest to the broad readership of EMBO Reports.

Referee #3:

The authors show that the inflammasome component NLRP1 is necessary but not sufficient for NGF deprivation-dependent axon degeneration in cultured sympathetic neurons and for infrapyramidal bundle pruning in mouse hippocampus. The studies in revised manuscript confirm that NLRP1-dependent activation of Casp1 and not Casp11 which distinguishes this from apoptotic death of neurons after NGF withdrawal. Overall, I think the authors have addressed my concerns and this will be a nice addition to the literature that distinguishes axon pruning mechanisms from apoptotic mechanisms.

All editorial and formatting issues were resolved by the authors.

Mohanish Deshmukh
University of North Carolina at Chapel Hill
United States

Dear Mohanish,

Thank you for implementing the final editorial formatting changes. I am very pleased to accept your manuscript for publication in the next available issue of EMBO reports. Thank you for your contribution to our journal.

Kind regards,

Martina
